# The Level of Stress and Coping Strategies in Patients with Multiple Sclerosis and Their Relationships with the Disease Course

**DOI:** 10.3390/jcm10173916

**Published:** 2021-08-30

**Authors:** Roman Kotas, Marta Nowakowska-Kotas, Sławomir Budrewicz, Anna Pokryszko-Dragan

**Affiliations:** 1Department of Psychiatry, Regional Specialist Hospital, ul. Iwaszkiewicza 5, 59-220 Legnica, Poland; romkot@interia.pl; 2Department of Neurology, Wroclaw Medical University, ul. Borowska 213, 50-556 Wrocław, Poland; slawomir.budrewicz@umed.wroc.pl (S.B.); anna.pokryszko-dragan@umed.wroc.pl (A.P.-D.)

**Keywords:** stress, multiple sclerosis, type-D personality, perceived stress, coping strategies

## Abstract

Objectives: Stress is supposed to be linked with a background of multiple sclerosis (MS) and the disease course. Design: The study aimed to assess the level of stress and coping strategies in MS patients within a year of follow-up and to investigate the relationships between these aspects and factors related—or not—to MS. Methods: In 65 patients with MS, the Perceived Stress Scale (PSS-10), Type D Scale (DS14) and Coping Orientations to Problems Experienced (COPE) were performed at baseline and after a year. Baseline PSS-10, DS-14 and COPE scores were analyzed with regard to demographics, MS duration, treatment, indices of disability and self-reported stressful events (SEs). Final PSS-10 and COPE results were analyzed with reference to MS activity and SE within a year of follow-up. Results: Initially, 67% of patients reported a moderate or high level of stress and 31% met Type-D personality criteria. Diverse coping strategies were preferred, most of which were problem-focused. The negative affectivity DS-14 subscore (NEG) was correlated with disability level. Non-health-related SEs were associated with higher PSS-10 and NEG scores. After a year, the mean PSS-10 score decreased, while COPE results did not change significantly. Non-health-related SEs were associated with a higher PSS-10 score and less frequent use of acceptance and humor strategies. Those with an active vs. stable MS course during the follow-up did not differ in terms of PSS-10 and COPE results. Conclusions: MS patients experienced an increased level of stress. No significant relationships were found between stress or coping and MS course within a year. Non-health-related factors affected measures of stress more than MS-related factors.

## 1. Introduction

Multiple sclerosis (MS) is a chronic disease of the central nervous system with a complex etiology and is associated mainly with immune system dysfunction. Multifocal damage to the brain and spinal cord, including inflammatory demyelination and axonal loss, results in a diversity of symptoms and signs of neurological deficit as well as emerging accumulating disability. Currently, two main phenotypes of MS are distinguished, relapsing and progressive, which are additionally modified by the presence or absence of activity and progression [1]. Furthermore, the course of the disease (both natural and modified by treatment) shows significant variability in the MS population. Thus, there is ongoing research on the role of factors influencing this course. Among these factors, stress is of great interest, especially in view of patient-related outcomes associated with their quality of life [2,3]. Several links have been established between MS etiology and biological mechanisms of stress, including hyperreactivity of the hypothalamic–pituitary–adrenal gland (HPA) axis, dysregulation of the autonomic system and subsequent modulation of the (auto)immune response [4,5,6]. These mechanisms can be additionally affected by the immunomodulating or immunosuppressive mechanisms of various types of disease-modifying therapies (DMT) used in MS.

On the other hand, MS itself may be a source of stress for patients due to its chronic and unpredictable course and emerging neurological deficits [7]. These complex links between MS and global distress (including depression, anxiety and overall emotional status) may create a so-called “vicious circle” [8,9]. The style of coping (mental and behavioral approaches undertaken by an individual to reduce any stress experienced) [10,11] is an important moderating factor and predictor of ultimate emotional distress [8,9]. The role of stressful events and coping in MS patients has already been investigated, but studies in this field have demonstrated a diversity of goals and methodologies, as well as inconsistent results [2,8].

The current study aimed to evaluate susceptibility to stress, its perceived level and coping strategies in patients with MS within a prospective annual observation. We also aimed to analyze the relationships between these aspects and MS-related variables and socio-demographic factors, including potentially stressful events.

## 2. Materials and Methods

The studied group comprised patients with relapsing–remitting (RR) MS, diagnosed according to McDonald’s criteria [12]. All of them were receiving DMT and were regularly consulted in the outpatient clinic at the Neurological Department with a documented course of the disease. Exclusion criteria were as follows: coexisting depression (based on the results of the Beck Depression Inventory [13]), severe cognitive dysfunction (based on Mini Mental State Examination [14]), which would prevent providing informed consent and completion of the questionnaires, and a relapse or introduction/switch of DMT within the preceding month.

Sixty-five patients were finally included in the study: 48 women and 17 men, with a mean age of 35.7 years (SD 7.3), all treated with interferon β. The following tests were performed in all the patients: the Perceived Stress Scale (PSS-10) [15] to evaluate the currently experienced level of stress, Type-D Scale (DS-14) [16] to identify a stress-prone D-type personality and Coping Orientations To Problems Experienced (COPE) [10] to determine preferred coping strategies (Appendix A). All the tests (in a standardized Polish version) were based on self-assessment questionnaires [17]. An additional questionnaire, formulated by the authors, was applied, including items on demographic factors (age, gender, marital status, occupational status, level of education) and questions about any stressful events (SEs) experienced within the preceding month and their type (related to health or not). The level of disability was assessed based on neurological examination using the Expanded Disability Status Scale (EDSS) [18]. The duration and course of the disease and treatment and the rate of disability accumulation in the Multiple Sclerosis Severity Scale (MSSS) [19] were based on medical records.

PSS-10 and COPE were repeated after 12 months, accompanied by the questionnaire on SEs experienced within this time. The data on the course of MS during this follow-up (relapses, change in the degree of disability and findings in magnetic resonance imaging (MRI)) were determined from medical records. MS activity within the analyzed period was evaluated with regard to clinical (relapses of the disease and/or accumulating disability) and radiological (new, especially active/contrast-enhanced lesions in MRI of the brain) measures. Those patients who had not experienced any of these were classified as having achieved a NEDA (No Evidence of Disease Activity) status [20].

The results of PSS-10 and COPE were compared between initial and final assessments. Relationships were sought between their initial scores and MS-related variables, sociodemographic factors and reported stressful events. Relationships between measures of stress and indices of disease activity (NEDA) within a year were also investigated.

Ethical approval was obtained from the local University Commission of Bioethics. All subjects provided informed consent prior to their inclusion in the study.

### Statistical Analysis

Statistical significance between means for independent groups was calculated by a one-way analysis of variance (ANOVA), alternatively using the non-parametrical Mann–Whitney U test when the variances in groups were heterogeneous (the Levene’s test determined the homogeneity of variance). The Chi-square test was used to calculate the statistical significance between frequencies in independent groups. The Wilcoxon’s non-parametric test (for continuous variables) and McNemar’s test (for discrete variables) were used for the assessment of dependent variables. The relation between two parameters was assessed using correlation analysis, and Spearman correlation coefficients were calculated. The Cronbach’s alpha was calculated to assess the reliability of scales. A vale of *p* < 0.05 was considered statistically significant. Statistical analysis was carried out using EPIINFO Ver. 7.1.1.14 (2 July 2013).

## 3. Results

### 3.1. Initial Assessment

Clinical characteristics of the studied group are presented in Table 1. Geographical and social characteristics are presented in Table 2. Table 3 shows the results of PSS-10 and DS-14. For COPE and PSS-10, a Cronbach’s alpha ranging from 0.704 to 0.794 was considered reliable. For DS-14, a Cronbach’s alpha of 0.228 was considered low. In PSS-10, 41% of patients reported high levels of stress, 26.2% reported moderate stress and 32.3% reported low stress. Among women, these proportions were 42%, 29%, and 29%; among men, they were 42%, 25%, and 33%, respectively, without significant sex differences. There was no correlation between PSS-10 results and the age of patients.

Type-D personality criteria were met by 20 patients (31%). There were no significant differences between the sexes in DS-14 results. DS-14 negative affectivity (NEG) subscores correlated significantly with age (*R* = 0.24, *p* < 0.05).

According to COPE results, MS patients reported the use of various coping strategies (Figure 1). Among the main categories of coping, problem-focused strategies were adopted most often and avoidant behaviors least often. The most frequently used strategies included planning, active coping and positive reinterpretation and growth; least frequently used were substance use, humor and behavioral disengagement. The only significant difference between the sexes was shown for substance use, which was more preferred by men (*p* < 0.05). Apart from the tendency (*p* = 0.09) towards the less frequent use of humor and planning in older patients, no significant relationships were found between COPE and age.

There was a significant correlation between PSS-10 and DS-14 NEG (*R* = 0.79, *p* < 0.001). Respondents with a type-D personality less frequently used the strategy of seeking emotional or social support (*p* < 0.01) than other respondents. No other significant relationships were found among the measures of stress.

#### 3.1.1. Measures of Stress and MS-Related Variables

There was a tendency (*R* = 0.21, *p* = 0.095) for higher PSS-10 results in patients with a longer MS duration. No significant correlations with PSS-10 were found for EDSS, MSSS or period of DMT use. A higher PSS-10 score (at the boundary of statistical significance; *p* = 0.053) was observed in those patients reporting adverse effects of DMT. There was no difference in PSS-10 results between those patients who had or had not experienced health-related SE.

NEG DS-14 subscores correlated significantly with EDSS (*R* = 0.22, *p* < 0.05). There were no other significant correlations between DS-14 results and MS-related variables or declared recent experiences of health-related SEs.

There were no statistically significant relationships between COPE results and MS-related factors. Observed trends included a negative correlation between the use of positive reinterpretation and MS duration (*R* = −0.21, *p* = 0.09) and more frequent use of restraint coping by those who had experienced health-related SEs (*R* = 0.22, *p* = 0.09).

#### 3.1.2. Measures of Stress and Other Factors

Patients that had experienced non-health-related SEs had significantly higher PSS-10 (15.9 pts vs. 15.7 pts, *p* < 0.05) and NEG DS-14 scores (*R* = 0.28, *p* < 0.05) than the others. Similar findings were noted for divorced persons compared to married and never-married respondents (PSS-10 *R* = 0.27, *p* < 0.05, NEG DS-14 *R* = 0.22, *p* < 0.05). A higher percentage of patients meeting the criteria for a type-D personality was found among pensioners in comparison with those who were professionally active or studying (chi^2^_3_ = 8.05, *p* < 0.05).

The burden of non-health-related SEs was correlated with less frequently reported turning to religion (*R* = −0.24, *p* < 0.05). Those patients who were studying/working showed a tendency for more frequent use of restraint coping (*R* = 0.21, *p* = 0.08) and behavioral disengagement (*R* = 0.21, *p* = 0.09). Unmarried and married subjects preferred positive reinterpretation and growth (*R* = −0.25, *p* < 0.05) and humor (*R* = 0.28, *p* < 0.05) more often than divorced and widowed respondents. Higher education levels correlated with a preference for problem-focused strategies (Cope A; *R* = 0.33; *p* < 0.01), planning (*R* = 0.27, *p* < 0.05) and restraint coping (*R* = 0.31; *p* < 0.05).

### 3.2. Assessment after a Year of Follow-Up

Sixty-three patients participated in the final assessment (two were lost during follow-up). In the analyzed period, 31 (49%) patients showed no features of disease activity (classified as NEDA 1), while in 32 (51%) patients, relapse and/or progression of disability and/or new lesions in brain MRI were observed (classified as NEDA 0) (Figure 2). Mean EDSS did not change after a year.

Subgroups NEDA 1 and NEDA 0 did not differ significantly in terms of age and gender, MS-related variables, and other factors determined in the initial assessment. Within a year of follow-up, in five patients, IFN β was switched to another DMT because of inefficacy or low tolerance, and in a further five, such a switch was planned.

Within a year, 18 subjects (13.8%) had experienced SEs related to MS and 26 (40%)-non-health-related SEs.

The mean PSS-10 score in the studied group after a year was significantly lower (17.2 pts vs. 15.6 pts, *p* < 0.05) than at the baseline. The preference for general or specific coping strategies had not changed significantly within a year, apart from the tendency for more frequent use of humor (*p* = 0.055) (Figure 3).

There were no significant differences between NEDA 1 and 0 subgroups in initial DS-14 and PSS-10 scores or the final PSS-10 score. Those patients whose PSS-10 score changed after a year did not differ significantly in terms of the disease activity from those with a stable score.

Patients classified as NEDA 0 changed their preferences for instrumental support strategies significantly more often than NEDA 1 (*p* < 0.05). There were no other relationships between COPE results and NEDA.

Patients who reported non-health-related SE within a year of follow-up had significantly higher final PSS-10 scores (18.1 pts vs. 13.8 pts, *p* < 0.05) and less frequently used the coping strategies of acceptance (2.4 vs. 2.8, *p* < 0.01) and humor (1.7 vs. 2.1, *p* < 0.05).

NEDA 1 and NEDA 0 subgroups did not differ in MS-related SE experienced before the initial assessment (7 vs. 8 subjects). However, the difference in this field was significant after a year of follow-up (0 vs. 18 subjects, *p* < 0.01). Non-health-related SE tended to be noted more often in the NEDA 0 subgroup in the initial assessment (17 vs. 9 subjects, *p* = 0.054), and after a year, there was a similar frequency in both subgroups (15 vs. 11 subjects).

## 4. Discussion

In the studied MS patients, almost 70% declared moderate or high levels of stress, but the mean PSS score did not differ significantly from Polish normative data [17]. Scheffer et al. [21] did not find differences in PSS scores and hair cortisol concentrations between MS patients and healthy controls, but both subgroups reported low overall levels of stress. In the study by Pritchard et al. [22], MS subjects presented with moderate mean PSS scores, but these were higher than those for the reference group of patients with cancer. In other studies comprising MS subjects, a diverse range of PSS scores has been obtained [23,24,25,26]. However, a similar wide range of PSS results has been reported in healthy populations [27,28,29,30] and in patients with inflammatory diseases [28,31,32]. Overall, these findings suggest substantial individual differences in perceived levels of stress, which are not straightforwardly determined by the experience of disease.

A Type-D personality is defined as a combination of negative affectivity (a tendency to experience negative emotions) and social inhibition (a tendency to inhibit self-expression in social interactions) [16]. It has been shown to be a risk factor for the incidence of and mortality from cardiovascular disorders [33,34], gastrointestinal diseases and neoplasms [35,36]. Approximately one-third of our MS patients met the criteria for a type-D personality, while the mean DS-14 result did not differ significantly from Polish normative data [17]. Although the Cronbach’s alpha value in our sample was low, the Polish version of DS-14 is regarded as a consistent and reliable tool for type-D personality assessment. In the literature, there is no clear evidence of a link between this personality type and MS, as both high and low incidence of type-D personality has been found in MS groups [37,38]. Type-D personality is associated with greater susceptibility to stress, with specific vasoactive reactions and cortisol excretion profiles as possible underlying mechanisms [39,40]. In our MS patients, no significant differences were found in the perceived levels of stress between those with and without a type-D personality. However, the PSS score was correlated with the results of the negative affectivity subscale of DS-14. The tendency to experience negative emotions in response to various situations might indeed result in being distressed [16] but not necessarily in defensive behavior.

Our MS patients preferred diverse coping strategies, with a predominance of problem-focused strategies and the lowest frequency of avoidant strategies. In contrast, other authors [41,42,43] have observed a trend towards passive and emotion-focused coping in MS subjects, although they used other tools for the evaluation of coping. Neither perceived level of stress nor type D personality were related to choice of coping strategies, which probably depended instead on the kind of stressors experienced and the individual situations of our MS subjects.

PSS and DS-14 scores in the studied group did not depend on age or gender, meaning that these factors could be ignored in further analysis. As for coping, trends were only observed for some preferences to increase and for others to decrease with age. The single difference between sexes concerned substance use, which was more preferred by men.

The studied group represented a relatively homogenous sample in terms of clinical characteristics. The perceived level of stress showed only a trend to increase with the duration of MS, with no other significant relationships with MS-related factors. Self-reported stressful MS-related events did not influence the level of stress either. It is worth highlighting that, due to exclusion criteria and the minimal MS duration of 12 months, recent diagnosis or relapse could not have been a source of stress. It was assumed that stressful events were associated with chronic symptoms or consequences of the disease. In other studies investigating perceived stress levels in MS patients [22,23,24,25,26], no relationships have been sought between PSS-10 and MS-related variables.

Of the DS-14 measures, only negative affectivity increased with the degree of disability in the studied group. Demirici et al. [38] found a correlation between EDSS and all DS-14 components, while other authors [44,45] have reported either links between disability and extraversion or no relationships between EDSS and personality.

Coping also seemed to be little affected by MS-related variables. Observed trends indicated a shift from problem-focused to avoidant strategies with increasing MS duration or health-related stressful events. Some authors [39,40] have made similar observations on avoidant coping strategies dependent on MS duration and disability level, while others [44,46] have not found such relationships.

Factors not related to MS showed more relevant relationships with indices of stress. Divorced persons reported a higher stress level, probably due to the stressful experience of divorce and its consequences. Higher PSS scores were obtained by those who had recently experienced non-health-related stressful events (as opposed to MS-related events). Other studies [47,48] suggest that MS patients may experience fewer life events than healthy controls but are more vulnerable to their adverse effects. Our patients were not asked to specify these events or rate their stressful impact. However, the quantification of this impact was provided by PSS, covering the same period (4 weeks), which requested the reiteration of stressful events.

Social-related factors also affected coping preferences. Single persons more often chose humor, positive reinterpretation and growth than divorced respondents. Higher education levels and professional activity were associated with a choice for problem-focused strategies but also substance use. Those who had recently experienced stressful events turned more often to substance use and less to religion-based coping. Taking into account the higher stress levels in this subgroup, the mentioned preferences might reflect a situational coping model and not general predispositions. These relationships seem to support the hypothesis that coping depends on resources (individual features, social status and interactions) and their availability [49].

Our study comprised a year of follow-up—a standard period used to evaluate DMT effectiveness—using the concept of NEDA [20]. In our study group, ca. 50% achieved NEDA within one year. The others experienced relapses and/or new MR lesions but without a progression in disability. No differences in initial MS-related variables were found between those who achieved NEDA after a year or not, which indicates difficulties in predicting response to treatment due to individual differences.

Assuming that personality type is a relatively stable set of traits and considering the low Cronbach’s alpha for DS-14 results, we did not re-evaluate type-D personalities after a year of follow-up, focusing on potential changes in levels of stress and coping profiles.

The perceived level of stress in the study group was significantly lower after a year of follow-up, presumably due to the effects of treatment (control over disease activity, no accumulating disability) and regular support from the medical setting. Other reports on MS subjects [22,23,25] also show a decrease in PSS but mostly as a short-term result of psychological interventions.

After a year, no significant change in coping preferences was noted in the study group. Lode et al. [50] also observed a tendency to preserve a stable coping model throughout the disease.

Our patients with a type-D personality or higher initial level of stress did not present with greater disease activity during the follow-up. Those who had not achieved NEDA more often reported MS-related stressful events within a year but without an increased perceived level of stress in the final assessment. Overall, our findings do not provide evidence either for stress triggering clinical/radiological disease activity or for an unfavorable course of MS as a source of stress.

There are numerous studies highlighting the impact of stress (incredibly intense and chronic or frequent) upon MS onset or exacerbations [6,48,51,52,53,54,55,56,57]. Biological and psychological factors have been proposed to contribute to these links [6,58]. However, some authors [42,59,60,61] have observed few or no relationships between stress and measures of MS activity, similar to our findings. Interpretation of these data is difficult due to the vast diversity of methodological approaches in the cited studies (especially with regard to the timeframe of recalled stressful events and the tools used for their reiteration) [8,9]. Moreover, contradictory results from studies conducted in similar settings (e.g., war zones) [62,63] suggest a substantial role of the individual perception of stress in the evaluation of its consequences.

The initially defined coping preferences in our patients did not predict MS activity within a year, and the adverse course of the disease during this time did not affect finally declared coping strategies either. Some authors [64,65] have suggested that a problem-focused approach may decrease the risk of exacerbations by lowering the level of stress, while emotion-focused and avoidant strategies badly influence patients’ adaptation to their situation [64,66]. Thus, some interventions are encouraged [67,68,69] due to their expected positive effect upon stress and adaptation levels through coping.

Our study comprises a prospective evaluation of various aspects of stress, using reliable measures, in a homogenous group of MS patients. The limitations include the moderate sample size (which prevented the performance of detailed multivariate analyses) and relatively short follow-up period, which might prevent the generalizability of the results. The findings might have also been biased by the subjective character and moderate internal consistency of the tools used for the evaluation of stress, as well as by a lack of quantification and detailed categorization of stressful events experienced by the subjects. However, our findings provide an encouraging background for further investigations including larger groups of MS patients and various clinical, psychological and socio-economic factors.

## 5. Conclusions

The majority of MS patients experience moderate to high levels of stress. Their preferred coping strategies are varied, with a predominance of problem-focused approaches. Non-disease-related factors affect the measures of stress more than MS-related variables. During the year of follow-up, the level of stress decreased while the coping profile remained stable. No significant impact of stress upon MS course was demonstrated within a year, nor for the influence of unfavorable MS course on the level of stress. These findings suggest the complex and individual nature of experiencing and managing stress, which deserves a personalized approach for MS patients.

## Figures and Tables

**Figure 1 jcm-10-03916-f001:**
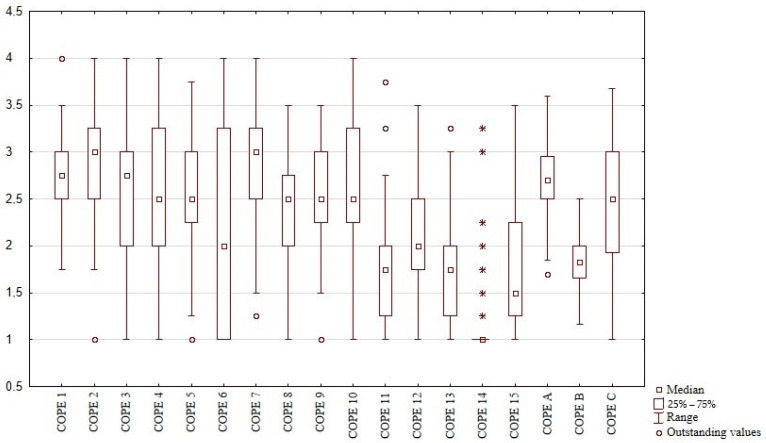
Frequency of coping strategies used in the study group according to COPE results. COPE 1—active coping, COPE 2–planning, COPE 3—seeking social support for instrumental reasons, COPE 4—seeking social support for emotional reasons, COPE 5—suppression of competing activities, COPE 6—turning to religion, COPE 7—positive reinterpretation and growth, COPE 8—restraint coping, COPE 9—acceptance, COPE 10—focus on and venting of emotions, COPE 11–denial, COPE 12—mental disengagement, COPE 13—behavioral disengagement, COPE 14—alcohol-drug disengagement, COPE 15—humor, COPE A—active coping, COPE B—avoidant behavior, COPE C—emotion-focused and seeking support strategy. * — individual scores (other subjects denied any use of alcohol).

**Figure 2 jcm-10-03916-f002:**
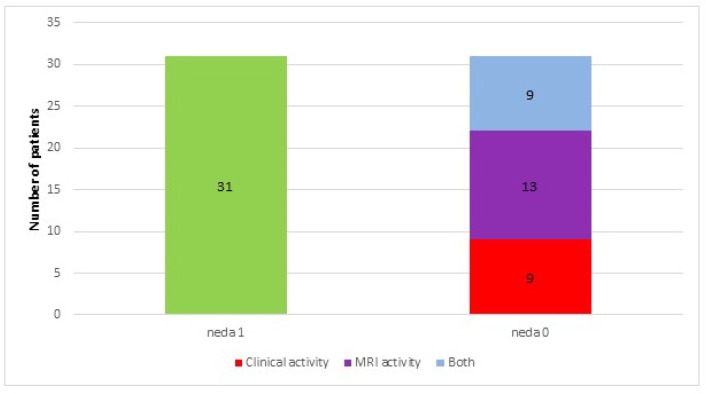
Numbers of patients who had no evidence of disease activity (NEDA 1) or presented with clinical and radiological activity measures (NEDA 0) within a year of follow-up.

**Figure 3 jcm-10-03916-f003:**
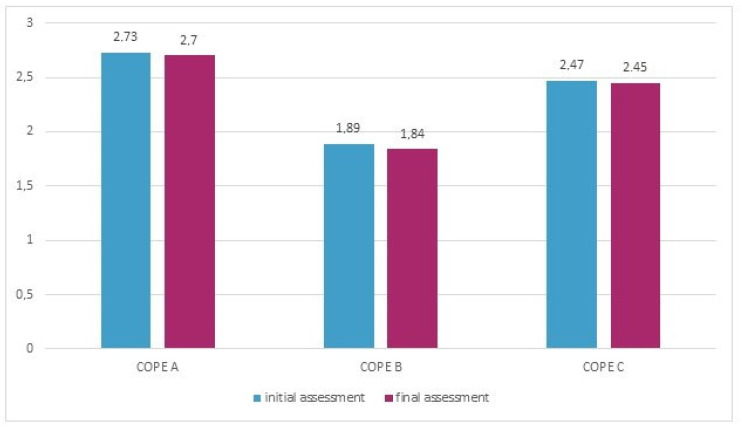
The use of general coping strategies (according to the COPE) in the study group in the initial and final assessments.

**Table 1 jcm-10-03916-t001:** Clinical characteristics of the group of MS patients under study (*n* = 65); EDSS—Expanded Disability Status Scale, MSSS—Multiple Sclerosis Severity Scale.

	Mean	SD	MIN	MAX	Median
Disease duration (years)	7.25	5.34	1	28	6
Duration of treatment	3.44	3.07	0.5	13	2.5
EDSS	2.05	5.4	1	6	2
MSSS	3.28	1.61	0.76	6.46	2.87

**Table 2 jcm-10-03916-t002:** Geographical (place of usual residence) and social characteristics of the group of MS patients (*n* = 65).

	Number	%
Place of residence		
Rural	22	33.8
Urban	43	66.2
Marital status		
Single	13	20
Married	40	61.5
Divorced	5	7.7
Other	7	10.8
Level of education		
Vocational	5	7.7
Secondary education	24	36.9
Higher education	36	55.4
Occupational status		
Studying	2	3
Working	52	80
Unnemployed	4	6.2
Pension/retiremet	7	10.8

**Table 3 jcm-10-03916-t003:** Results of Perceived Stress Scale (PSS-10) and Type-D Scale (DS-14) in the group under study (*n* = 65); NEG—negative affectivity, SI—social inhibition subscales of DS-14.

	Mean	SD	MIN	MAX	Median
PSS-10	16.9	6.7	3.0	28	17
DS-14 NEG	12.4	7.1	0	27	12
DS-14 SI	8.97	6.38	0	25	8

## Data Availability

The data that support the findings of this study are available from the corresponding author, [MNK], upon reasonable request.

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
