# Peer review of "The Level of Stress and Coping Strategies in Patients with Multiple Sclerosis and Their Relationships with the Disease Course"

_jcm, 2021, doi:10.3390/jcm10173916_

Round 1
Reviewer 1 Report
I have only one comment regarding the introduction. On line 35 of the first page the authors refer to the disease courses by speaking of an active (relapsing-remitting) and progressive form. This is wrong, because there are two courses: relapsing and progressive. Both can be active or inactive, with progression and without progression. See the work of Lublin F 2014 for a correction of the sentence.
Author Response
Thank you very much for your favourable review - the MS classification has of course been taken into account, with reference to Lublin F.D, Neurology 2014
Reviewer 2 Report
General comments: The manuscript reports the results of a prospective study that assessed perceived stress and coping strategies (and their clinical and sociodemographic correlates) in patients with relapsing-remitting multiple sclerosis (MS) at baseline (n=65) and one year later (n=63). At baseline, 67% of patients reported important levels of stress, 31% met Type-D personality criteria, and mostly employed problem-focused coping. One year later, perceived stress significantly decreased while coping strategies remained unchanged. Correlation analysis revealed significant correlations at both time points. At baseline, negative affectivity subscore (of type-D personality), perceived stress and less use of the coping strategies acceptance and humors, were associated with non-health-related stressful events. At one year, perceived stress and a less employment of acceptance and humor coping strategies were associated with non-health-related stressful events. The authors suggest the contribution of non-health related stressful events, rather than disease characteristics, to stress and coping in patients with MS. The results are interesting, but the manuscript has several limitations some of which have been already addressed by the authors. Please see below for a detailed review.
Editing: The manuscript would benefit from editing. Here are some examples:
- In the abstract, it is important to clarify the employment of ‘referred to’.
- ‘disease’ would be more appropriate than 'disorder' when referring to MS (l.31).
- The symbol of chi square needs to be corrected in the statistical analysis section.
Introduction: It is advisable to discuss the three types of MS (relapsing-remitting, primary progressive and secondary progressive).
Methods:
- It is important to clarify what screening/clinical tools were used to rule of coexisting depression and severe cognitive dysfunction ruled out.
- It is important to clarify why personality D assessment was only done at baseline (e.g., stable traits?)
- It is important to state if the correlation analysis was corrected for multiple comparisons. If not, this limitation merits to be discussed since it can increase the chance of type I error.
- Regarding the used scales, it would be helpful to mention the adopted cutoffs to classify mild, moderate and severe stress levels and having or not personality D criteria.
- The study is exclusively based on subjective scales and the internal consistency of these scales (alpha Cronbach) merits to be tested. This is particularly interesting in the case of coping strategies since some of them have been found to have low internal consistency in some studies, which might limit the interpretation of some results.
Results:
- Adding descriptive statistics of marital status, occupational status, level of education is helpful.
- It is important to provide information about the questionnaire used to assess the stressful events.
- The results section is sometimes difficult to follow (alternation between descriptive statistics, groups comparisons and correlation analysis, and in some cases, there is a difficulty to understand the type of the statistical relationship (group comparison or correlation analysis). Therefore, it is advisable that the authors start by discussing for each timepoint separately (baseline and at year 1) the cohort’s descriptive characteristics (clinical data, sociodemographic, data stress/coping/personality D) followed by correlation analysis. Group comparison between baseline and year 1 could be displayed in a separate section. In addition, the sentences could be paraphrased to allow understanding in a more concrete manner the employed statistical tests.
- It is advisable to standardize the presentation of p values (p<0.05, p<0.01, p<0.001; there is no need to mention p<0.000001). The authors can also present p values up to 3 decimals to provide more precision (e.g., p=0.xxx).
- It would be helpful to define some acronyms in tables’ legends (i.e., EDSS and MSSS in table 1)
- In figures, it would be easier for readers to replace COPE 1-14 and COPE A-C with their names.
Discussion: This section would benefit from a further discussion of the limitations.
References: It would be of importance to cite a work by Lazarus and Folkman regarding coping.
Author Response
Thank you very much for such an insightful analysis of our work.
In response to individual comments:
1) the abstract has been modified (lines 15 and 17)
2) methods - the description of methods has been clarified. Both in the text (lines 65 - 67 and 74 - 78) and in the appendix.
3) statistics - multiple comparisons were not performed - hence these calculations have not been modified now. The paragraph on statistics has been clarified and the chi notation corrected. The notation P was also standardised, as suggested by the reviewer
4) statistics - Cronbach's for DS-14 and PSS-10 test was calculated, which was taken into account in lines 112-114 and in discussion 299-301
5) Methods - coexisting depression has been ruled out by clinical examination (psychiatrist) and Beck Depression Inventory
6) Table two was added - to add information on geographical, socio-economic data of patients as suggested by the reviewer. However, definitions and courses of multiple sclerosis were not elaborated due to limited space.
7) The description of the results has been changed to make it easier for readers
8) The table legends have been supplemented. In Figure 1, the addition of the full names of the strategies resulted in a complete lack of legibility of the figure - the description under the figure has been expanded as a substitute.
9) The limitations in discussion has been further highlighted (lines 300-301 and 333-338)
10) Lazarus R and Folkman S has been cited (position number 11)
Thank you again for your insight into the work and I hope that the additions and explanations are satisfactory
Round 2
Reviewer 2 Report
The referee would like to thank the authors for the attention they paid to the raised comments and suggestions. A few issues remain to be addressed:
DS-14 Cronbach alpha was considered low - 0.228 and negative which might limit the validity of this measure in this paper. However, a negative Cronbach's alpha indicates inconsistent coding or a mixture of items measuring different dimensions, leading to negative inter-item correlations. Therefore, calculating Cronbach alpha for the two dimensions of DS-14 might help solving the issue as the internal consistency of the Polish version of the DS14 was good with Cronbach's alpha of 0.86 for negative affectivity and 0.84 for social inhibition (10.2478/s10059-009-0029-8).
The correlation analysis was not corrected for multiple comparisons which might increase the risk of type 1 error (false positivity). It is acceptable not to apply correction, but the issue needs to be addressed among the limitations.
Editing comments:
-Table 2 decimals should be expressed as periods ‘.’ and not commas ‘,’
-References style should be adopted in p. 8 (Artemiadis et al., 2012; Sanaeinasab et al., 2017; Senders, Bourdette, Hanes, Yadav, & 409 Shinto, 2014).
Author Response
Thanks to the Reviewer for this careful analysis of our text.
Referring to the comments:
1) DS-14 Cronbach alpha was considered low......
There has been a mistaken understanding of the dash as a minus sign. We have put the number in brackets in line 181 for better readability.
By this method, Cronbach's alpha is further down in value, but not so much. Of course, at the suggestion of the reviewer, we have additionally included that the DS-14 tool itself was tested and validated under Polish conditions (appendix and lines 337-338)
2) The correlation analysis was not corrected ....
Our statistical analysis was mainly based on comparisons rather than correlations, but of course, the moderate size of the group prevented wider use of multivariate analysis with appropriate adjustments. Hence, this was further highlighted within the limitations (line 477).
3) Table 2 decimals should be expressed as periods ‘.’ and not commas ‘,’
Thank you for catching this editorial error
4) References style should be adopted in p. 8 ...
One more time: thank you for catching this editorial error
Thanks to the Reviewer very much, for his/her contribution so far to the improvement of this work and in case of further doubts and comments - I am at the disposal